# SIR-2.1 integrates metabolic homeostasis with the reproductive neuromuscular excitability in early aging male *Caenorhabditis elegans*

**Xiaoyan Guo[1], L René García[1,2]***

[1]Department of Biology, Texas A&M University, College Station, United States;
[2]Howard Hughes Medical Institute, Texas A&M University, Texas, United States

**Abstract** The decline of aging *C. elegans* male's mating behavior is correlated with the increased excitability of the cholinergic circuitry that executes copulation. In this study, we show that the mating circuits' functional durability depends on the metabolic regulator SIR-2.1, a NAD+-dependent histone deacetylase. Aging *sir-2.1(0)* males display accelerated mating behavior decline due to premature hyperexcitability of cholinergic circuits used for intromission and ejaculation. In *sir-2.1(0)* males, the hypercontraction of the spicule-associated muscles pinch the vas deferens opening, thus blocking sperm release. The hyperexcitability is aggravated by reactive oxygen species (ROS). Our genetic, pharmacological, and behavioral analyses suggest that in *sir-2.1(0)* and older wild-type males, enhanced catabolic enzymes expression, coupled with the reduced expression of ROS-scavengers contribute to the behavioral decline. However, as a compensatory response to reduce altered catabolism/ROS production, anabolic enzymes expression levels are also increased, resulting in higher gluconeogenesis and lipid synthesis.

***For correspondence:** rgarcia@bio.tamu.edu

**Competing interests:** The authors declare that no competing interests exist.

## Introduction

Although lifespan is well studied in the model organism *Caenorhabditis elegans* (**Kenyon, 2010**), the aging process per se is still under intense research (**Jin, 2010**). For example, behavioral and/or mental ability starts to decline during early aging, prior to any dramatic structural or morphological dysfunction (**Salthouse, 2004**; **Guo et al., 2012**). The decline could be due to physiological alterations toward a non-optimal state, which leads to the failure of proper behavioral execution. Therefore, it is urgent to uncover the molecular mechanism(s) underlying the physiological changes, so that certain actions could prevent or postpone the non-optimal modifications that occur during aging.

Our previous studies used *C. elegans* males to establish a behavioral model for studying physiological alterations that occur during early aging (**Guo et al., 2012**). The spicule intromission motor step, which occurs during *C. elegans* male copulation, is sensitive to changes that occur during early aging (**Garcia et al., 2001**; **Garcia and Sternberg, 2003**). Through pharmacological, optogenetics, and genetic analyses, we found a correlation between the increased excitability of the spicule intromission circuit and the decline of mating at early adulthood (**Guo et al., 2012**). However, the molecular mechanisms underlying this correlation were unknown.

Caloric restriction is an effective way to extend lifespan (**Hursting et al., 2003**). In *C. elegans*, caloric deprivation such as transient starvation for 3 to 18 hr during early adulthood can reduce excitability in the spicule intromission circuitry and prolong the mating potency of wild type (**LeBoeuf et al., 2011**). We showed that the effect of transient starvation is partially mediated by UNC-43/CaMKII and up-regulation of the *unc-103* and *egl-2*-encoded ERG-like K+ channels (**LeBoeuf et al., 2011**; **Guo et al., 2012**). Although males with mutations in both K+ channels abrogate most of the

**eLife digest** Although the signs of aging are clear to us all, precisely why we age is less well understood. One possibility is that as cells use oxygen to fuel the breakdown of large molecules into smaller ones to release energy, they also generate by-products called reactive oxygen species that can damage DNA. As we get older, this damage gets worse. Consistent with this idea, it has been shown that a reduced calorie intake can reduce oxidative damage in certain species, in addition to extending lifespan.

Many experiments on aging have been performed on worms belonging to the species *C. elegans*. Male worms of this species live for an average of 11–12 days, but begin to show signs of aging—for example, a reduced ability to mate—as early as day 3 of their adult lives. Now, Guo and García have revealed that a protein called SIR-2.1, which regulates metabolism in worms, also helps to protect the animals from the effects of aging.

Male worms in which the gene for this protein has been 'knocked out' have a normal lifespan, but show signs of aging earlier than normal males. They are also more susceptible to the damaging effects of reactive oxygen species, suggesting that SIR-2.1 may offer protection against oxidative damage. Indeed, levels of ATP—the molecule used to move energy around inside cells—are increased in knockout worms. This suggests that certain metabolic processes and the production of reactive oxygen species, are increased in the knockout worms, which speeds up the aging process.

While the link between metabolism and aging is well known, the work of Guo and García offers insights into some of the molecular mechanisms that may form the basis of this relationship.

beneficial effects of transient starvation, we still observed a small increase in the mating ability of aged double-mutant males, suggesting that other mechanisms are applied by transient starvation to improve mating (*Guo et al., 2012*). One possibility is that metabolic alteration, induced by transient starvation, compensates for the physiological changes that occur during aging. The connection between metabolic change and physiological status might be the generation of reactive oxygen species (ROS). First, metabolic status determines the oxidative stress burden: ROS is the byproduct of oxidative phosphorylation (*Murphy, 2009*). Second, emerging evidence shows that ROS modulates the excitability of neuromuscular system through modification of ion channels (*Taglialatela et al., 1997*; *Cai and Sesti, 2009*; *Aggarwal and Makielski, 2013*). However, there are few comprehensive in vivo studies, which link ROS-mediated physiological changes to complex behavior alteration.

To address this, we investigated how a mutation in the *C. elegans* metabolism regulator, SIR-2.1, alters behavior. SIR-2.1 is an ortholog of yeast SIR2 (*Tissenbaum and Guarente, 2001*). Yeast SIR2, a $NAD^+$-dependent histone deacetylase, with a role in chromatin regulation, can silence ribosome DNA expression and recombination (*Gottlieb and Esposito, 1989*). When overexpressed, *sir2* extends yeast lifespan, whereas deletion of the gene shortens the lifespan by 50% (*Kaeberlein et al., 1999*). Although there is controversy on whether overexpression of invertebrate *sir2* ortholog extends lifespan (*Burnett et al., 2011*; *Viswanathan and Guarente, 2011*), sirtuin family proteins have been shown to regulate glucose and fat metabolism (*Houtkooper et al., 2012*). For example, the *C. elegans* SIR-2.1 inhibits lipid synthesis during fasting (*Walker et al., 2010*). In addition, sirtuin proteins mediate an oxidative stress response by regulating antioxidant gene expression through transcriptional factors such as FOXO in a 14-3-3-dependent manner (*Berdichevsky et al., 2006*; *Webster et al., 2012*; *Merksamer et al., 2013*). Overall, sirtuin proteins could be involved in age-related diseases, such as type II diabetes and neurodegenerative diseases (*Satoh et al., 2011*; *Houtkooper et al., 2012*). Taken together, it is possible that sirtuin proteins regulate the cell's metabolic status and physiology, which further alters the coordination of specific behaviors.

In this study, we used *C. elegans* male mating behavior to study the molecular and physiological alterations underlying the behavioral decline that occur during early aging. We found that wild-type males require SIR-2.1 to maintain mating potency, and *sir-2.1* mutant males show a premature decline in copulation behavior, consistent with oxidative stress-induced hyperexcitability of their mating circuit. We propose that in *sir-2.1(0)* males, enhanced glycolysis/fatty acid oxidation, coupled with a compromised anti-stress system, contribute to premature mating decline. However, in mutant and aged wild-type males, pyruvate carboxylase and phosphenolpyruvate carboxykinase are up-regulated,

likely as a compensatory mechanism to shunt excess pyruvate from glycolysis and the TCA cycle to lipid biosynthesis, gluconeogenesis, and glyceroneogenesis.

## Results

### SIR-2.1 maintains male mating during early aging

Previously, we reported that *C. elegans* male mating behavior deteriorates during early aging. N2 and *him-5(e1490) C. elegans* (henceforth, will be referred to as wild type) males' mating capability begins to decline at day 3 of their adulthood, although their median lifespan is 11–12 days (*Guo et al., 2012*). We demonstrated that transient starvation of young males can extend their mating span, partially through up-regulation of *ether-a-go-go* K+ channels (*LeBoeuf et al., 2011*); however, our data also suggested that transient starvation can improve mating through additional mechanisms (*Guo et al., 2012*). Considering that metabolism is altered in food-deprived males (*Tan et al., 2011*), we tested whether perturbing the histone deacetylase metabolic regulator, *sir-2.1*, affects the functional span of copulation behavior in fed and transiently starved/re-fed males.

In adult hermaphrodites, animals that lack *sir-2.1* have increased intestinal lipids (*Walker et al., 2010*), a phenotype opposite of starved animals. Likewise, we found that 1-day-old *sir-2.1(ok434)* null *(0)* males also contain more lipids than wild type (*Figure 1A*). In addition, we observed that 2-day-old wild-type males have more fat (*Figure 1A*). Thus, we asked if *sir-2.1(0)* males might have altered mating due to metabolic dysregulation. Allowing the males to mate for 5 hr, we found that well-fed aging *sir-2.1(0)* males' mating ability drops prematurely, compared to wild-type males (*Figure 1B* (i)). Even under unlimited mating conditions, the mating potency of 2-day-old *sir-2.1(0)* drops to 42% (p<0.0001, n = 47) (*Figure 1B* (ii)).

We then asked if transient starvation can suppress the mating defect in *sir-2.1(0)*. To do so, we starved *sir-2.1(0)* males for ~20 hr from L4, and conducted a 5 hr mating potency assay. Transient starvation improved mating of 2-day-old *sir-2.1(0)* males from 13% to 75% (p<0.0001, *Figure 1C*), but at day 3, the mating potency of transiently starved *sir-2.1(0)* males was still lower than wild type. Thus, similar to wild type, the metabolic alteration and/or up-regulated EAG K+ channel functions caused by starvation alleviate some of the dysfunction caused by the *sir-2.1* deletion. However, the mutant's reduced mating potencies between day 1 and 3 under both conditions indicate that SIR-2.1 contributes to the functionality of the mating circuits during this period.

To confirm that premature mating deterioration in *sir-2.1(0)* males is caused by the *ok434* allele, we introduced into *sir-2.1(0)* animals a rescuing transgene containing the *sir-2.1* endogenous promoter driving the *sir-2.1* genomic sequence fused to *yfp*. The extrachromosomal expression of *sir-2.1* significantly improved the mating potency of 2-day-old *sir-2.1(0)* males from 26% to 75% (p<0.0001, *Figure 1D*). *sir-2.1* is expressed broadly in *C. elegans* (*Bamps et al., 2009*), thus we further conducted tissue-specific rescue assays, but found that none of the tissue-specific promoters driving the expression of *sir-2.1*, including neuronal, muscle, and intestinal promoters can rescue the premature mating decline (*Figure1—figure supplement 1A*), suggesting that *sir-2.1* is required in multiple tissues to maintain male mating.

To exclude the possibility that *sir-2.1(0)* mating deficiency at day 2 is due to a shorter lifespan, we conducted a lifespan assay and found that *sir-2.1(0)* males lived as long as wild type (*Figure 1—figure supplement 1B*). Another possibility for the mating deterioration is morphological deformities of the sex musculature. However, in 2-day-old *sir-2.1(0)* males expressing a functional *yfp:act-1* transgene (*Figure 1—figure supplement 1C*), we did not observe any obvious muscle fiber disorganization, which normally occurs in 8-day-old wild-type males (*Guo et al., 2012*). Although we did not inspect neural morphology, published studies showed that neural morphology does not change in *C. elegans* during aging (*Herndon et al., 2002*).

Another potential explanation for the lower mating efficiency is that sperm activity in 2-day-old *sir-2.1(0)* males is defective. *C. elegans* male sperm are stored in the seminal vesicle as a non-activated form and become activated after transfer into a hermaphrodite. Regulated by proteases, individual sperm goes through a morphological change to form a pseudopod. This pseudopod provides mobility for the sperm to fertilize the hermaphrodite oocyte (*Smith and Stanfield, 2011*). To test whether the low mating potency of 2-day-old *sir-2.1(0)* males is due to failure in sperm activation, we did an in vitro sperm activation assay and found that similar to wild type, 92.0 ± 4.3% of sperms from 2-day-old *sir-2.1(0)* can be artificially activated by pronase (*Figure 1—figure supplement 1D*).Taken together,

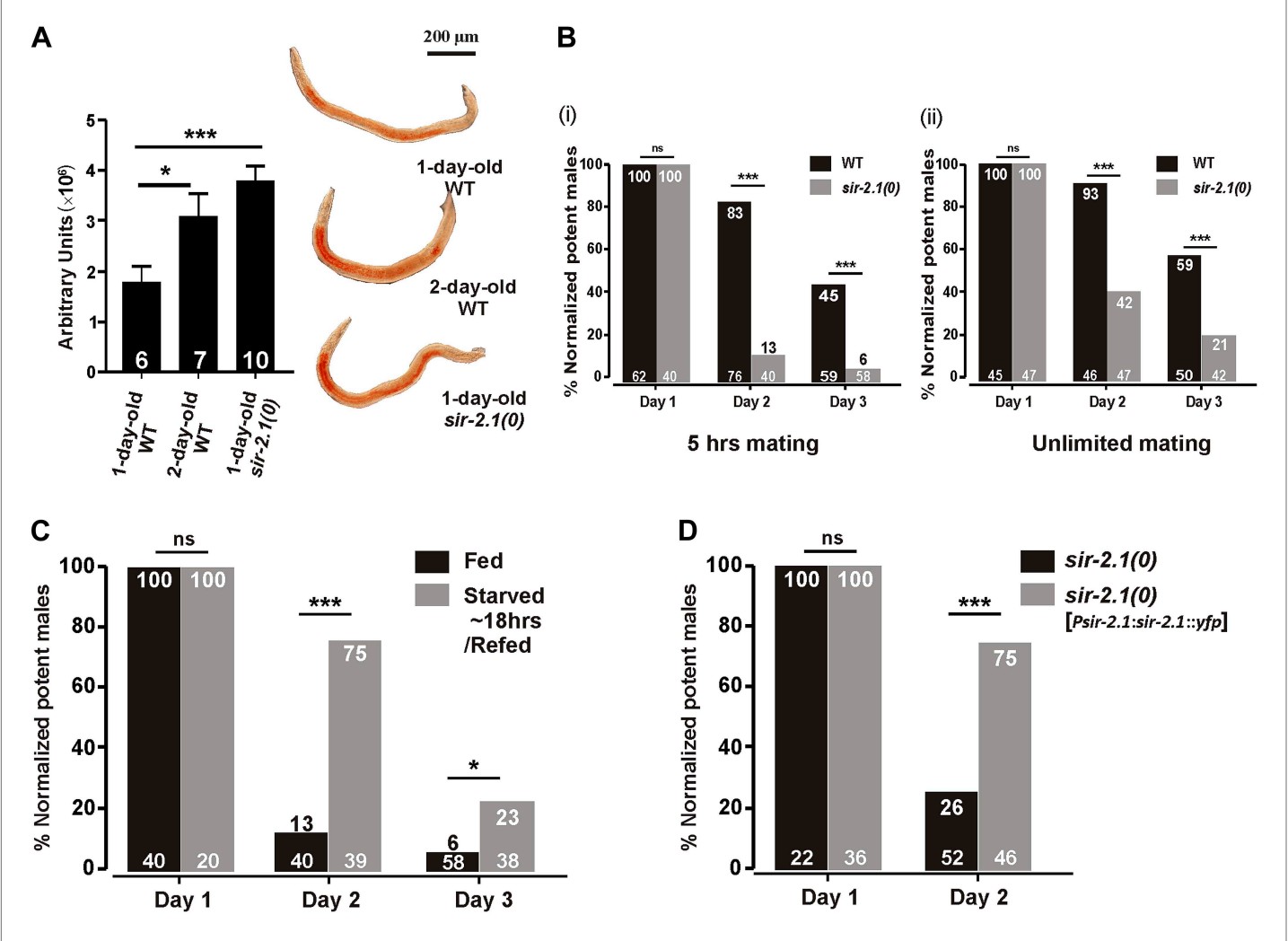

**Figure 1**. *sir-2.1(0)* males have altered lipid content and their mating ability deteriorates prematurely. (**A**) 1-day-old *sir-2.1(0)* and 2-day-old wild-type males have more lipid content than 1-day-old wild type. Left: quantification of fat staining, mean ± SEM, unpaired t-test. Right: representative images of fat staining. (**B**) Mating potency of wild-type and *sir-2.1(0)* males. Copulations were allowed to occur for 5 hr (i) or for an unlimited time (ii). The number of males in each assay is listed at the bottom of each bar. The numerical percentage of wild-type males that mated on day 1 was normalized to 100%. The normalization factor was then applied to the other experimental conditions. The normalized percentages for each day are listed on the top. Fisher's exact test was used to compare the mating potency prior to normalization. (**C**) Transient starvation reduces *sir-2.1(0)* mating deficiency. (**D**) Mating potency of *sir-2.1(0)* and rescued strain *sir-2.1(0); rgEX399 [Psir-2.1:sir-2.1::yfp]*. ns, not significant. Asterisks *, ** and *** indicate the p<0.05, 0.01, and 0.0001 in this paper, respectively.

The following figure supplements are available for figure 1:

**Figure supplement 1**. (**A**) Tissue specific expression of *sir-2.1* does not rescue the reduced mating potency of *sir-2.1(0)* males at day 2.

we speculate that the premature mating decline in *sir-2.1(0)* is due to physiological changes, rather than the structural degeneration of either neuromuscular circuits or sperm.

## *sir-2.1(0)* males mating circuit becomes more excitable

We showed that wild-type mating deterioration at day 3 is correlated with an increased excitability in the mating circuitry (*Guo et al., 2012*). Hence, we hypothesized that *sir-2.1(0)* mating decline might also be due to a premature increase in the cellular excitability. To test this, we used two acetylcholine (ACh) agonists, levemisole (LEV) and arecoline (ARE) to determine the responses of wild-type and *sir-2.1(0)* males at multiple ages. In the male spicule intromission circuit, LEV binds to ionotropic

ACh receptors, whereas ARE is a nonselective ACh agonist (*Liu et al., 2007*; *Correa et al., 2012*). Activation of ACh receptors depolarizes the male's neurons and muscles, and ultimately causes sex muscle contractions; as a result, males protrude their copulatory spicules. We found that at day 1, *sir-2.1(0)* and wild-type males had similar response to a sub-threshold effective concentration of ARE (50 μM) (*Figure 2A* (i)). However, 2-day-old *sir-2.1(0)* males were more sensitive to agonist stimulation and required significantly less time to respond (*Figure 2A* (ii)). Additionally, 2-day-old *sir-2.1(0)* males were more sensitive to sub-threshold LEV stimulation. 58% *sir-2.1(0)* compared to 35% of wild type protracted their spicules in 500 nM LEV ($p < 0.05$, n > 30) (*Figure 2B* (ii)). These results indicate that the loss of *sir-2.1* in males alters the spicule intromission circuit's excitability during early aging.

## Hyperexcitability leads to an ejaculation defect

To address how hyperexcitability disrupts copulation, we observed the mating behavior of 2-day-old *sir-2.1(0)* and wild-type males. We found that 2-day-old *sir-2.1(0)* males performed most of the mating steps similarly to the wild-type control (*Figure 2— figures supplement 1A,B,C,D*). Although 2-day-old *sir-2.1(0)* males can effectively insert their spicules, a significant number of them failed to transfer sperm into the hermaphrodite (*Figure 2C*). Upon spicule insertion, sperm moved out from the seminal vesicle and traveled through the vas deferens; however, they remained stuck in the vas deferens and did not drain through the cloacal opening (*Videos 1, 2 and 3*). Even the exceptional *sir-2.1(0)* males that successfully ejaculated, transferred less sperm and produced fewer progeny (*Figure 2D*).

The male copulatory spicules are attached to three sets of sex muscles: the retractor, protractor, and anal depressor muscles. Contraction of the protractor muscles leads to spicules insertion into the vulva (*Garcia et al., 2001*). During the normal ejaculation step of mating behavior, after spicule penetration, the posterior gubernaculum erector and retractor muscles contract, presumably to pull the proctodeum posteriorly, so that sperm can drain from the vas deferens (*Figure 3—figure supplement 1*) (*Liu et al., 2007*). In *sir-2.1(0)* males, we speculated that after spicule insertion, the abnormal increased cell excitability causes the spicule protractor and anal depressor muscles to hypercontract, which during sperm transfer, would pinch close the vas deferens opening. To test this, we imaged the $Ca^{2+}$ in the spicule-associated dorsal protractor and anal depressor muscles (region-of-interest, ROI, indicated in *Figure 3A,B*), by expressing G-CaMP3 in these sex muscles of both *sir-2.1(0)* and wild-type males (*Tian et al., 2009*; *Guo et al., 2012*). During the mating behavior of 2-day-old wild-type males, the G-CaMP3 ΔF/F0 increased to 129.0 ± 32.5% (n = 5) at the time of spicule insertion, and the $Ca^{2+}$ signal started to decline to 86.7 ± 30.8% during the 10-s period after spicule insertion (*Figure 3A* and *Video 4*). This indicates that the spicule protractor muscles partially relax after spicule insertion. However, in 2-day-old *sir-2.1(0)* males, ΔF/F0 increased up to 204.3 ± 97.5% (n = 5) upon spicule insertion. Unlike wild type, $Ca^{2+}$ transients did not decrease as much, and the ΔF/F0 fluctuated at about 129.8 ± 33.0% (*Figure 3B* and *Video 5*). The sustained higher $Ca^{2+}$ levels in 2-day-old *sir-2.1(0)* males suggest that spicule protractor and anal depressor muscles are hypercontracted and pinch the vas deferens opening, thus blocking sperm release.

## Reactive oxygen species leads to the mating deterioration

*C. elegans* hermaphrodite studies showed that SIR-2.1 promotes the expression of antioxidant genes through its association with the FOXO/DAF-16 transcription factor (*Berdichevsky et al., 2006*). *sir-2.1(0)* hermaphrodites are more sensitive to stresses such as reactive oxygen species (ROS) (*Rizki et al., 2011*). Therefore, we asked if ROS-induced damage contributes to the premature mating deterioration. We confirmed that similar to hermaphrodites, *sir-2.1(0)* males are also more sensitive to paraquat, a ROS-generator. Mutant males are less viable in 10 mM paraquat after 24 hr; 89% of *sir-2.1(0)* males survived, compared to 99% of wild type ($p < 0.01$, n > 100) (*Figure 4A*). When exposure time reached 48 hr, the difference between two strains became more obvious, 39% of wild-type males survived, compared to 4% of *sir-2.1(0)* ($p < 0.001$) (*Figure 4A*).

Since *sir-2.1(0)* males are more sensitive to oxidative stress, we hypothesized that during aging, accumulated ROS from metabolism might contribute to the decreased mating efficiency and to the increased excitability of the spicule intromission circuit. To test this, we grew wild-type males on plates containing 1 mM paraquat from late L4 to adult and assayed their mating ability and genital muscle excitability. After exposure to paraquat for 24 hr, males showed significant decline in mating potency (*Figure 4B*). Additionally, these males also displayed increased genital muscle sensitivity to the day 1

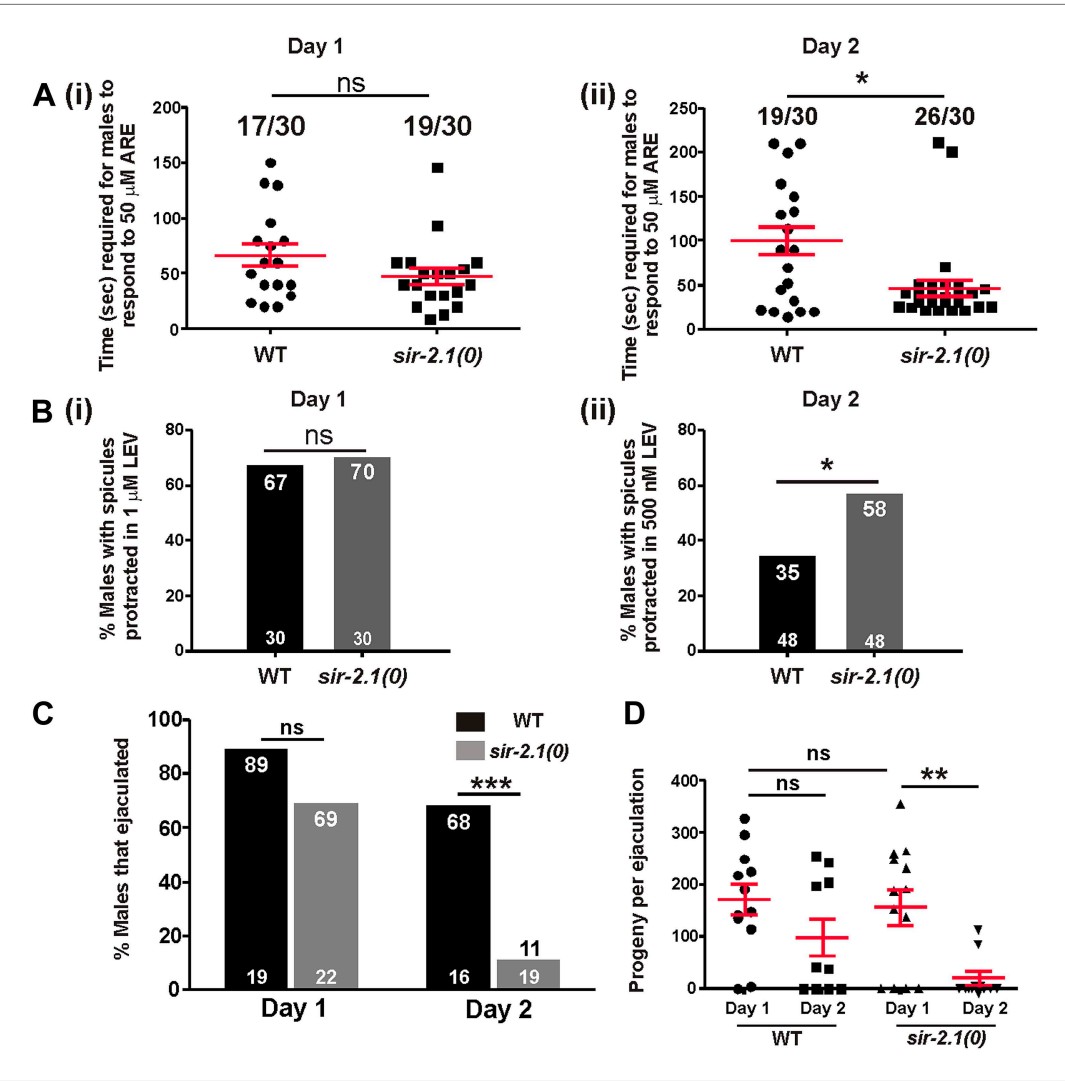

**Figure 2**. *sir-2.1(0)* males' sex circuitry becomes more excitable during aging, and those males display ejaculation defects. (**A**) 1-day-old wild-type and *sir-2.1(0)* males (n = 30) have similar response to the ACh agonist arecoline (ARE). The time required for those males to protrude their spicules out in 50 μM ARE solution are not significantly different (i) (unpaired t-test), whereas 2-day-old *sir-2.1(0)* males require significantly less time to respond to ARE (ii) (unpaired t-test). Mean and SEM are indicated. (**B**) 1-day-old wild-type and *sir-2.1(0)* males (n = 30) have similar response to the ACh agonist levamisole (LEV) (i). However, 2-day-old *sir-2.1(0)* males are more sensitive to LEV (ii). (Fisher's exact test). (**C**, **D**) 2-day-old *sir-2.1(0)* males have an ejaculation defect. (**C**) The percentages of 2-day-old wild-type and *sir-2.1(0)* males that ejaculated during copulation. (Fisher's exact test). (**D**) The numbers of cross progeny produced by individual 2-day-old wild-type and *sir-2.1(0)* with *unc-64(e240)* hermaphrodites. Mean and SEM are indicated (unpaired t-test).

The following figure supplements are available for figure 2:

**Figure supplement 1**. (A) A cartoon illustration of *C. elegans* male mating behavior.

---

$EC_{50}$ concentration (1 μM) of LEV. 56% of wild type protracted their spicules; however, 83% males exposed to paraquat responded to the ACh agonist (p<0.05, n = 54) (*Figure 4C*).

To further test if ROS contributes to the copulation decline, we supplemented the males' media with the antioxidant N-acetyl-cysteine (NAC) (*Schulz et al., 2007*), and asked if NAC can delay genital muscle excitability changes and improve fertility. Indeed, when we exposed wild-type and *sir-2.1(0)* males to NAC from L4 to adulthood day 3 and adulthood day 2, respectively, the antioxidant decreased

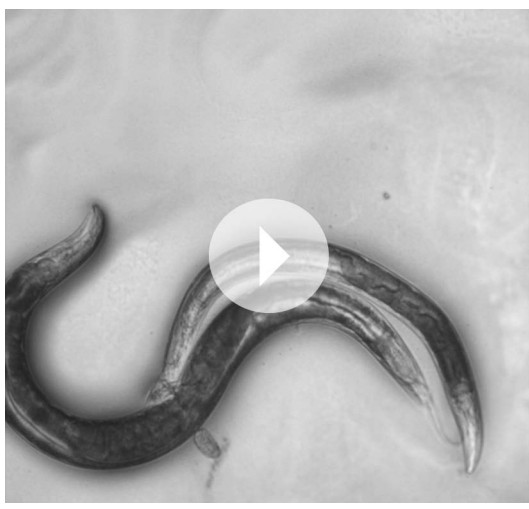

**Video 1**. Wild-type male's ejaculation.

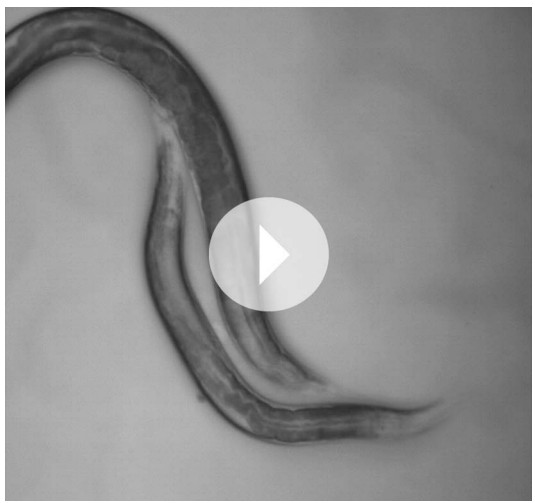

**Video 2**. 2-day-old *sir-2.1(0)* male's ejaculation.

the males' sensitivity to 100 nM LEV (the $EC_{50}$ concentration for older males [*Guo et al., 2012*]) (*Figure 4D,E*) and increased their mating potency (*Figure 4F,G*). These results are consistent with the idea that ROS contributes to the behavioral decline.

## *sir-2.1(0)* males might have altered metabolism and reduced ROS scavenge capability

Next, we asked why 2-day-old *sir-2.1(0)* males are more sensitive to ROS. Metabolism as a major source of endogenous ROS stress might contribute to behavioral decline. SIR-2.1's role in regulating metabolic processes has not been well described in *C. elegans* hermaphrodites, and scarcely in males. Therefore, we compared the expression levels of 55 genes involved in multiple metabolic processes including: glycolysis, gluconeogenesis/glyceroneogenesis, citrate acid cycle, glyoxylate cycle, fatty acid metabolism and electron transport chain (ETC)/oxidative phosphorylation (OXPHOS) (*Castelein et al., 2008*) between age-matched *sir-2.1(0)* and wild-type males. Out of the 55 genes we surveyed, 17 showed statistically significant changes (*Figure 5*); the information for all the genes is tabulated in the *Supplementary file 1*.

Through real-time PCR analyses, we found that mRNAs encoding key enzymes involved in the initiation of glycolysis (hexokinase [F14B4.2] and glucose-6-phosphate isomerase [Y87G2A.8]) and fatty acid oxidation (fatty acid acyl-CoA synthetase [C46F4.2]) were up-regulated in 1-day-old *sir-2.1(0)* and 2-day-old wild-type males, relative to 1-day-old wild type (*Figure 5A,B*). This is consistent with the published observation that hexokinase is also up-regulated in the whole body of conditional *sirt1* knock-out mice (*Gomes et al., 2013*). Most enzymes involved in the TCA cycle did not change in their levels (*Supplementary file 1*). In contrast, expression of ETC/OXPHOS components (*cco-1*, W09C5.8) was reduced in *sir-2.1(0)* (*Figure 5F*). Other genes that were significantly up-regulated include key anabolic enzymes like fatty acid desaturase (*fat-5,6,7*), pyruvate carboxylase (PC) (*pyc-1*) and phosphoenolpyruvate carboxykinase (PEPCK) (*pck-1* and *pck-2*), isocytrate lysase (*icl-1*) and aconitase-cytosol (*aco-1*), which are important for fatty acid biosynthesis, gluconeogenesis, glyceroneogensis, and glyoxylate cycle (*Figure 5D,E*) (*Yang et al., 2009*). Consistent with this, hepatic cells without *sirt1* also have an up-regualtion of PEPCK gene expression (*Wang et al., 2011*). Fatty acid desaturase plays a critical role in lipid/triglyceride biosynthesis (*Van Gilst et al., 2005*; *Flowers and Ntambi, 2008*). PC catalyzes the carboxylation of pyruvate to oxaloacetate (OAA), the first step that shunts pyruvate from glycolysis to gluconeogenesis/glyceroneogenesis. Alternatively, OAA, an intermediate of TCA, can be converted to phosphoenolpyruvate (PEP) by PEPCK directly inside the mitochondrion or transported and converted to PEP by PEPCK in cytosol (*Figure 6A*). The up-regulation of fatty acid desaturase is consistent with the increased lipid staining in *sir-2.1(0)* males (*Figure 1A*).

To confirm if the changes in mRNA levels of those metabolic enzymes reflect functional alterations in the metabolic processes, we also measured ATP, glucose and glycogen accumulation in wild-type and *sir-2.1(0)* males. Consistent with increased expression levels of glycolysis and fatty acid oxidation

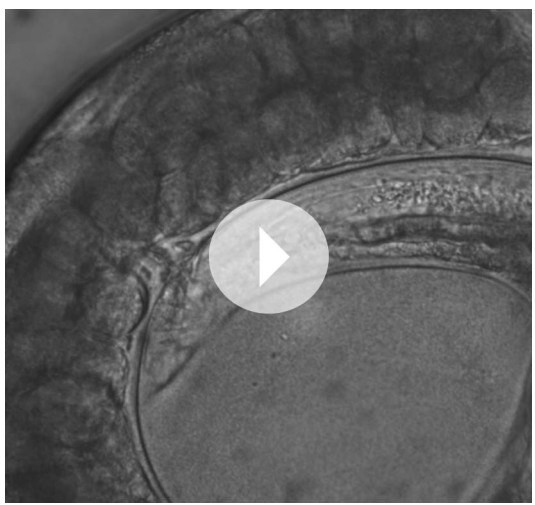

**Video 3**. 2-day-old *sir-2.1(0)* male's ejaculation.

genes, *sir-2.1(0)* males produced significantly more ATP at day 1. At day 2, wild-type ATP levels increased to match the level of *sir-2.1(0)* males. But at day 3, *sir-2.1(0)* males again accumulated more ATP (*Figure 6B*). These data suggest that *sir-2.1(0)* and older wild-type males might have an enhanced catabolism, consistent with the potential to generate more ROS.

Based on metabolic roles of PEPCK (*Yang et al., 2009*), up-regulation of *pck* genes could lead to excess glucose/glycogen in *sir-2.1(0)* males. Although similar amounts of glucose were detected in *sir-2.1(0)* and wild-type males (*Figure 6—figure supplement 1*), more glycogen was synthesized in the mutant (*Figure 6C*). In addition to gluconeogenesis, PEPCK catalysis of OAA to PEP is also a key step for the synthesis of glycerol-3-phosphate, which is used in triglyceride biosynthesis (*Figure 6A*) (*Nye et al., 2008*). Indeed, males lacking functional *pck-2*, but not *pck-1* have less lipid content (*Figure 6D*).

Additionally, *sir-2.1(0)* males that lack *pck-2,* but not *pck-1*, showed reduced lipid staining (*Figure 6D*), indicating that *pck-2* is necessary for the up-regulation of fat synthesis in *sir-2.1(0)*. Taking together the real-time PCR results, accumulation of metabolic products and hypersensitivity to paraquat, we propose that in *sir-2.1(0)* males, glycolysis and fatty acid oxidation are up-regulated to provide excessive NADH to the electron transport chain. Since we also measured reduced expression of ETC components cytochrome c oxidase, more ROS might be generated via electron leak (*Lee et al., 2010*). However, we hypothesized that the enhanced expression of enzymes involved in anabolic processes might be a suboptimal self-compensatory mechanism to shunt excess pyruvate from being oxidized in the TCA cycle.

To test if the up-regulation of *pck* genes *in sir-2.1(0)* is a compensatory response, we assayed the mating potency of *pck-1(0)* and *pck-2(0)* single mutants and *sir-2.1(0); pck-1(0)* and *pck-2(0); sir-2.1(0)* double mutants. At day 1, *sir-2.1(0)* and *pck-2(0)* males mated comparable to wild type (*Figure 6E*). However at day 2, the potency of *pck-2(0)* males started to decline similarly to *sir-2.1(0)*. In contrast, for males containing both *sir-2.1(0)* and *pck-2(0)*, their mating potency dropped at day 1 (*Figure 6E*). This indicates that without *pck-2*, males that contain or lack *sir-2.1* display accelerated behavioral decline. Similar to the requirement for functional *pck-2*, males that lack *sir-2.1* also needed *pck-1* to maintain their mating potency at day 1; however, *pck-1* was not required for *sir-2.1(+)* males to mate efficiently at day 2 (*Figure 6F*).

Next, we reasoned that if excessive glycolysis contributes to the behavioral deterioration, artificially adding extra glucose to the males' media could accelerate their mating decline. To test this, we grew males on UV-killed-OP50 NGM plates supplemented with 2% glucose, from hatched larvae up to the adult age prior to behavioral decline, which is day 1 or day 2, for *sir 2.1(0)* and wild-type males, respectively. We found that the glucose reduced mating potency of 1-day-old *sir-2.1(0),* but not 2-day-old wild type (*Figure 6G*), indicating that wild type can cope with the extra glucose better than *sir-2.1(0)*.

We hypothesized that unlike wild type, *sir-2.1(0)* males cannot efficiently respond to the oxidative stress generated by the enhanced catabolism. To test this, we used qRT-PCR to measure the mRNA levels of antioxidant genes: superoxide dismutase (*sod-1, 2, 3, 4* and *5*), catalase (*ctl-1, 2*), and glutathione transferase (*gst-10* and *gsto-1*) relative to 1-day-old wild-type males (*Figure 7*). As expected, the expression of *sod-1, sod-5, gst-10* and *gsto-1* was reduced in 1-day-old *sir-2.1(0)* and 2-day-old wild type (*Figure 7*). For *sod-2*, day 1 expression was also reduced in *sir-2.1(0)*, but this gene's expression increased in both wild type and *sir-2.1(0)* at day 2, possibly a stress response. For *sod-3* and *ctl-2*, their day 1 expression was similar in both strains; however at day 2, *sod-3* expression became higher and *ctl-2* expression became lower in mutants. Finally, *sir-2.1(0)* males displayed an increased *ctl-1* expression at day 1, which is also reported in antioxidant-compromised *daf-16(0)* mutant. The enhanced expression of *ctl-1* is considered as an adaptive response (*Yanase et al., 2002*). These results indicate

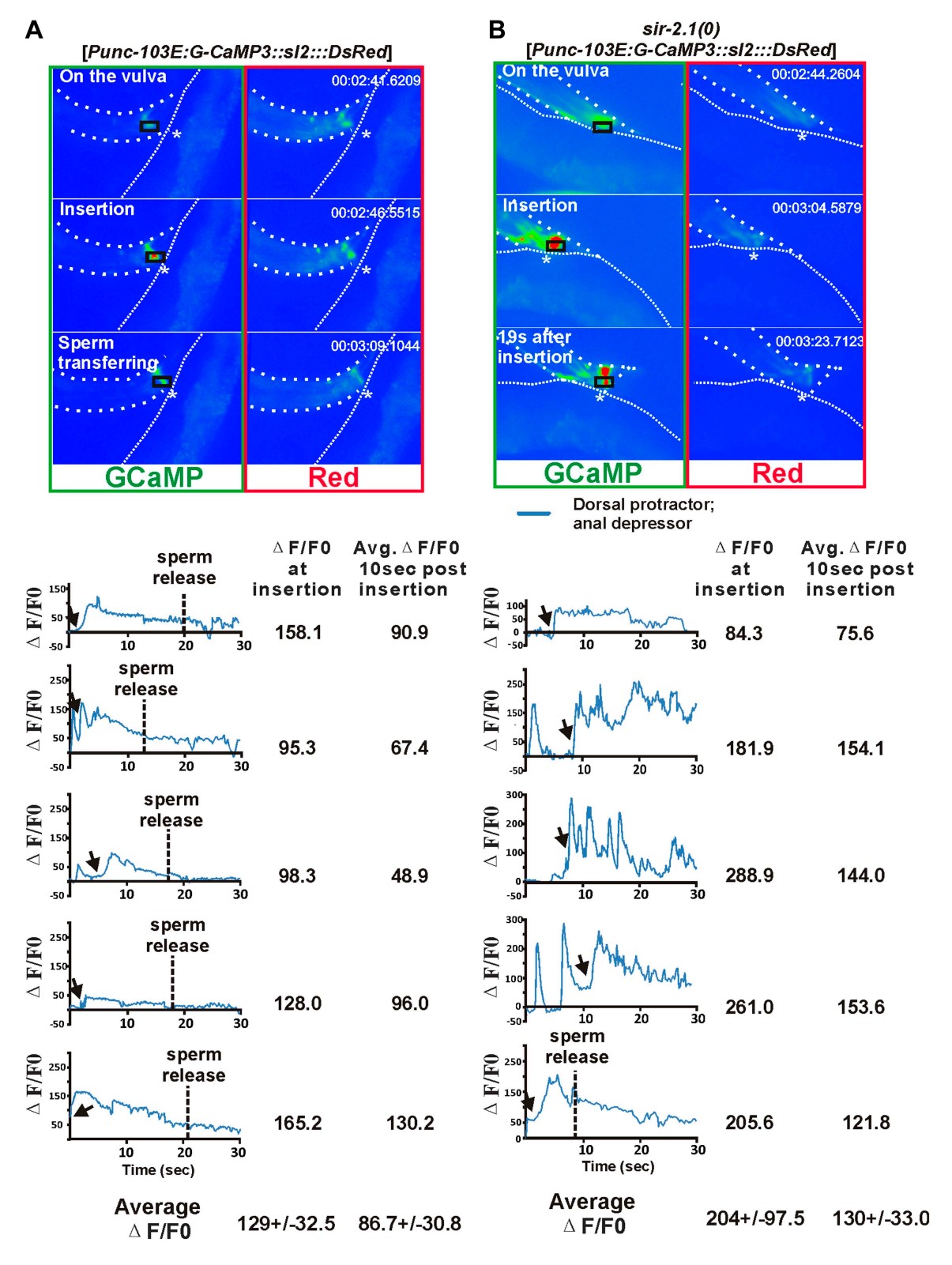

Figure 3. Ca²⁺ imaging of spicule-associated muscles in mating males. Pseudo-colored images of Ca²⁺ in the spicule muscles of 2-day-old wild-type and *sir-2.1(0)* males during mating (**A**) and (**B**) are representative frames to show the Ca²⁺ levels of the spicule-associated muscles during spicule insertion attempts, penetration and the start of sperm transfer (~10 s after insertion for wild type) or 19 s after insertion (for *sir-2.1(0)* males). The asterisks indicate

*Figure 3. Continued on next page*

*Figure 3. Continued*

the hermaphrodite vulva. Below the images, the Ca²⁺ transients in the protractor and anal depressor muscles (indicated by the black rectangle in **A** and **B**) are plotted for 5 individual wild-type (**A**) and *sir-2.1(0)* males (**B**), respectively.

The following figure supplements are available for figure 3:

**Figure supplement 1**. A cartoon illustrating the contractile changes of the spicule-associated muscles during intromission and ejaculation behaviors of a 2-day-old wild-type and *sir-2.1(0)* male, respectively.

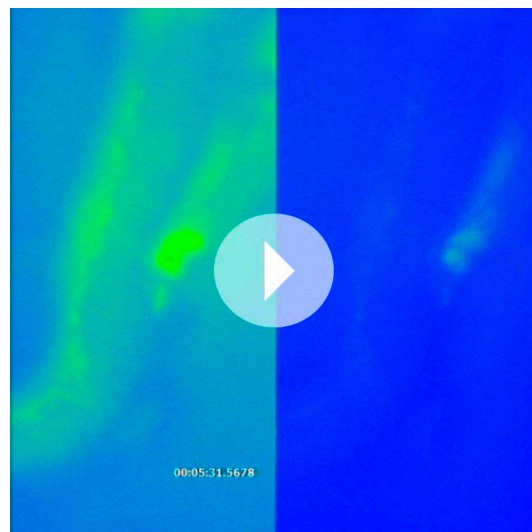

**Video 4**. Ca2⁺ transient in a 2-day-old wild-type male.

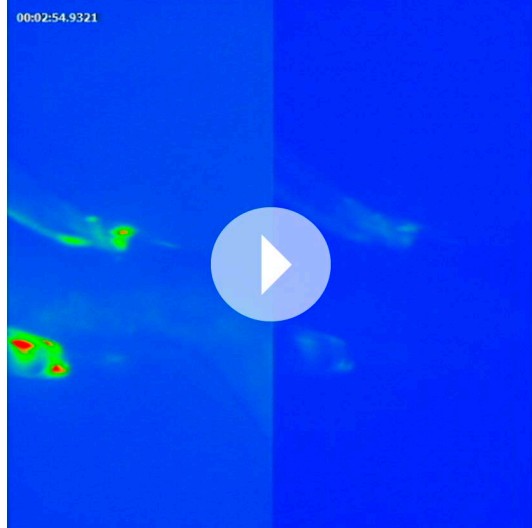

**Video 5**. Ca2⁺ transient in a 2-day-old *sir-2.1(0)* male.

that in addition to a potentially altered metabolism, which could generate excessive ROS, *sir-2.1(0)* males might also have a comprised antioxidant response, which is consistent with their hypersensitivity to excessive glucose intake (*Figure 6G*) and to the ROS generator (*Figure 4A*).

## Nicotinamide delays the deterioration of male mating behavior

Based on the above results, one could hypothesize that increasing SIR-2.1 expression or activity might delay mating deterioration during aging. However, we found that transgenic overexpression of *sir-2.1* does not improve the mating potency of 3-day-old wild type (*Figure 8A*). It is unlikely that the fusion with YFP disrupts SIR-2.1 function, because the same transgene can rescue the *sir-2.1(0)* phenotype. Thus, we speculate that up to a point, the expression level of *sir-2.1* is not rate limiting for SIR-2.1 activity during early aging. However, one could also speculate that the normal endogenous levels of NAD⁺ limit the function of SIR-2.1. To test this, we grew males in the presence of the NAD⁺ precursor nicotinamide (NAM) at 200 μM concentration (*Houtkooper et al., 2010*), and then conducted the mating potency. Indeed, NAM exposure significantly improves 3-day-old wild-type mating potency, but not 2-day-old *sir-2.1(0)* males (*Figure 8B,C*). This result is consistent with the idea that excess NAD⁺ might stimulate SIR-2.1 activity. But additionally, excess NAD⁺ might also reduce ROS production by relaxing the demand of oxidizing NADH back to NAD⁺; as a corollary to this possibility, the lack of excess NAM to positively affect the *sir-2.1(0)* male's behavior might be aggravated by the abnormally high expression of catabolic enzymes in the mutant males. To further test if overexpressing SIR-2.1 activity can promote mating behavior in older males, we exposed 3-day-old transgenic SIR-2.1 over-expressed males with exogenous NAM, but found that excess SIR-2.1 does not amplify the positive effect of NAM (*Figure 8D*). Thus, we cannot exclude the possibility that NAM or possibly NAD⁺ additionally promotes behavioral extension through mechanisms parallel to SIR-2.1 activity.

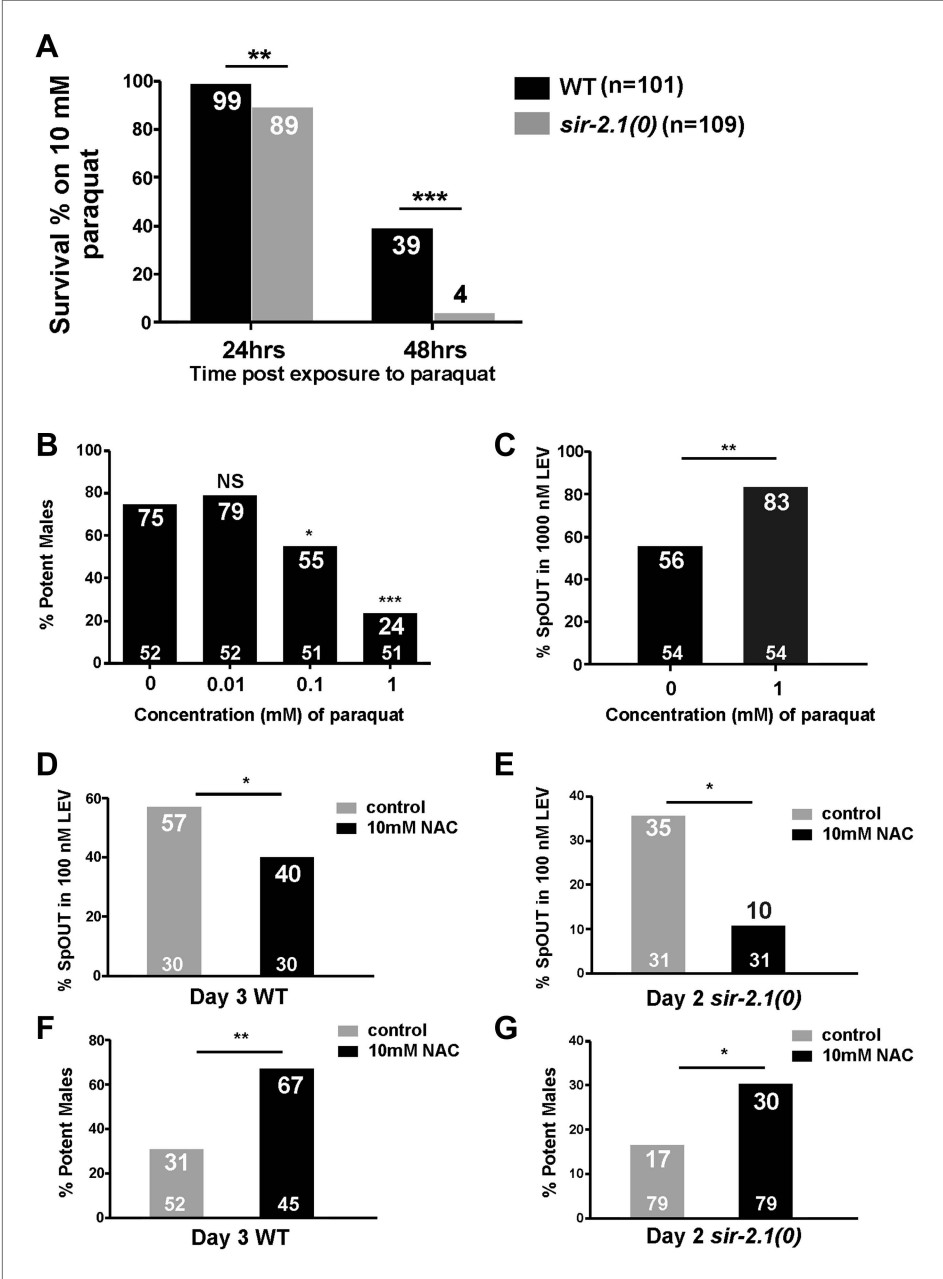

**Figure 4**. ROS contributes to the mating deterioration. (**A**) Survival rates of wild-type and *sir-2.1(0)* males on NGM containing 10 mM paraquat at 24 hr and 48 hr post paraquat exposure. (**B**) Mating potency of 1-day-old wild-type males exposed to 0.01, 0.1, and 1 mM paraquat. (**C**) The percentages of males with their spicules protruding out (SpOUT) in response to 1 μM levamisole (LEV) after treatment with 1 mM paraquat. (**D**–**G**) Exposing males to N-acetyl-cystine (NAC) improves mating. The percentages of 3-day-old wild-type (**D**) and 2-day-old *sir-2.1(0)* (**E**) males that protrude their spicules out in response to 100 nM LEV after NAC exposure. Mating potency of 3-day-old wild-type (**F**) and 2-day-old *sir-2.1(0)* (**G**) males after NAC exposure (Fisher's exact test).

## Discussion

Behavioral decline at advanced age can be attributed to morphological degeneration, such as neuronal death and muscle sarcopenia (*Herndon et al., 2002*; *Glenn et al., 2004*). However, during early aging, prior to any significant cellular decay, physiological changes that affect neuronal and muscular functionality can contribute to the reduction of behavioral coordination (*Salthouse, 2004*). Male

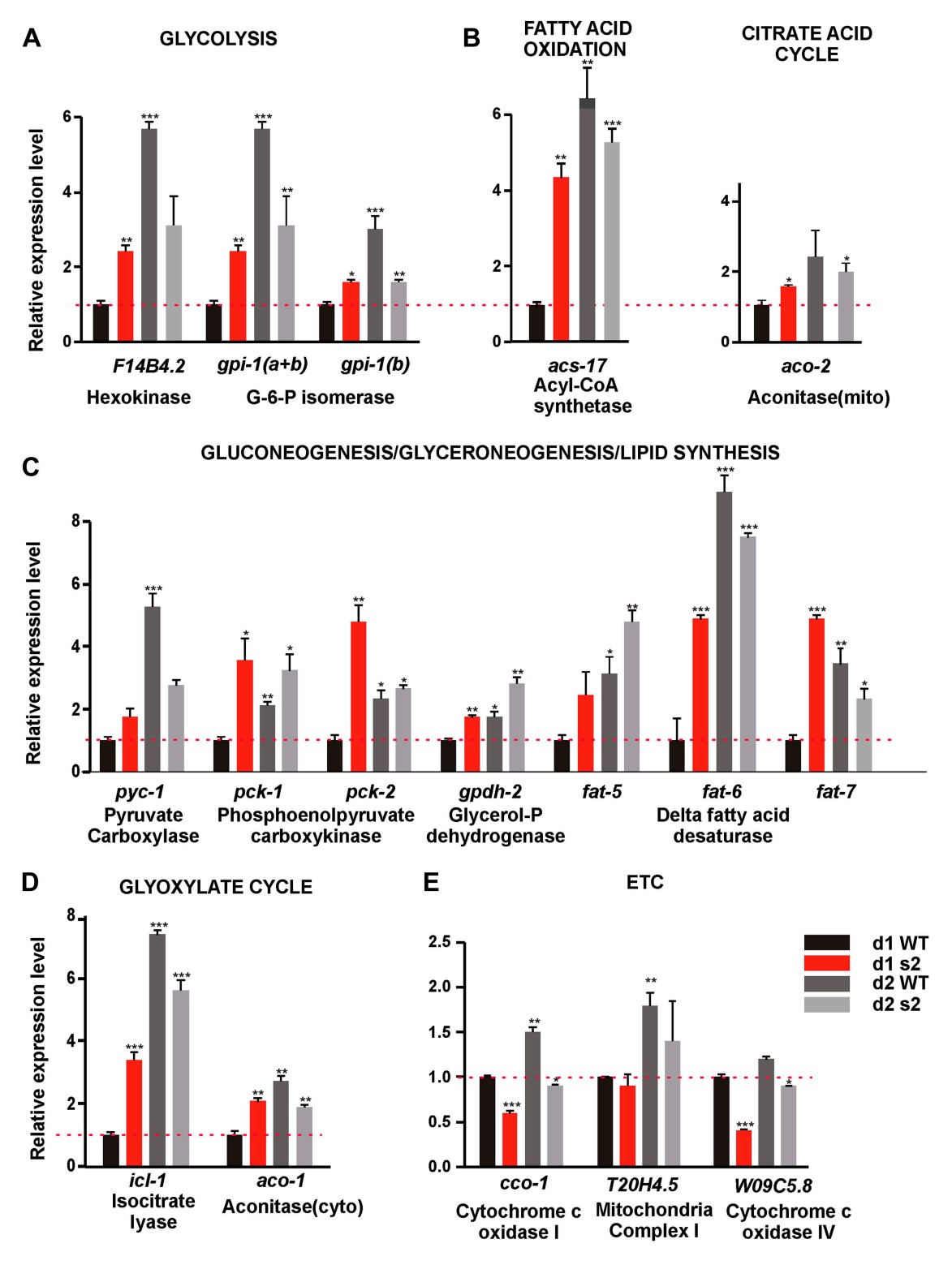

**Figure 5**. *sir-2.1(0)* males have altered expression of metabolic genes. Relative mRNA expression level of genes involved in metabolic processes such as glycolysis (**A**), TCA cycle (**B**), fatty acid oxidation(**C**), Gluconeogenesis/glyceroneogenesis/lipid synthesis (**D**), Glyoxylate cycle (**E**), and ETC (**F**) in 2-day-old wild type, 1-day-old, and 2-day-old sir-2.1(0) males relative to 1-day-old wild type. d1 WT refers to day1 wild type; d2 WT refers to day 2 wild type; d1 s2 refers to day1 *sir-2.1(0)*; d2 s2 refers to day 2 *sir-2.1(0)* (unpaired t-test compared to 1-day-old wild type).

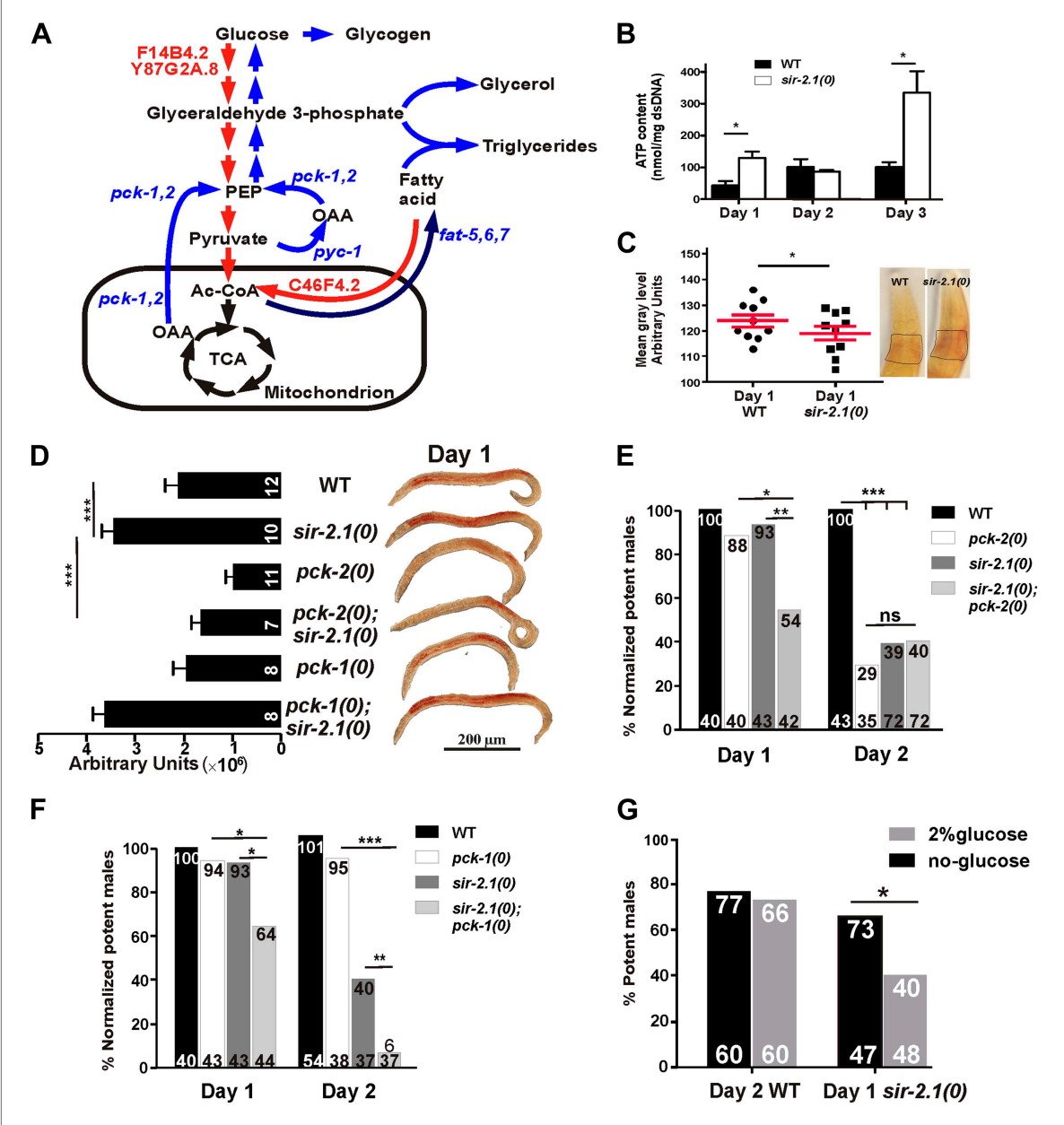

**Figure 6**. *sir-2.1(0)* males might have enhanced metabolism. (**A**) Schematic illustration of main metabolic enzymes which have altered expression in *sir-2.1(0)* males. Red arrows indicate catabolic pathways. Blue arrows indicate anabolic pathways. (**B**) ATP content measured in 1, 2 and 3-day-old wild-type and *sir-2.1(0)* males. (**C**) Glycogen staining in 1-day-old wild type and *sir-2.1(0)*. The glycogen staining level is quantified by measuring the mean gray level of the ROI indicated on the top right corner. The mean gray level is inversely correlated with the iodine stain. (**D**) Oil Red O staining of wild type and mutant *C. elegans* males. (**E**) *sir-2.1(+)* and *sir-2.1(0)* need *pck-2* to maintain their mating at day 2 and day 1 respectively. All percentages of mating potency are normalized to that of 1-day-old wild-type male. (**F**) *sir-2.1(0)* requires *pck-1* to maintain their mating at day 1 and day 2, while *sir-2.1(+)* males do not need *pck-2* to maintain their mating at either day 1 or day 2. (**G**) 2% glucose reduces 1-day-old *sir-2.1(0)* mating potency, but not 2-day-old wild-type males (Fisher's exact test).

The following figure supplements are available for figure 6:

**Figure supplement 1**. The level of glucose content is similar between 1-day-old *sir-2.1(0)* and wild-type males.

mating in *C. elegans* is a complex behavior that requires coordination of multiple motor systems to impregnate the hermaphrodite (*Liu et al., 2011*; *Correa et al., 2012*). Previously, we showed that male mating behavior declines significantly during early aging. Wild type cannot mate well at

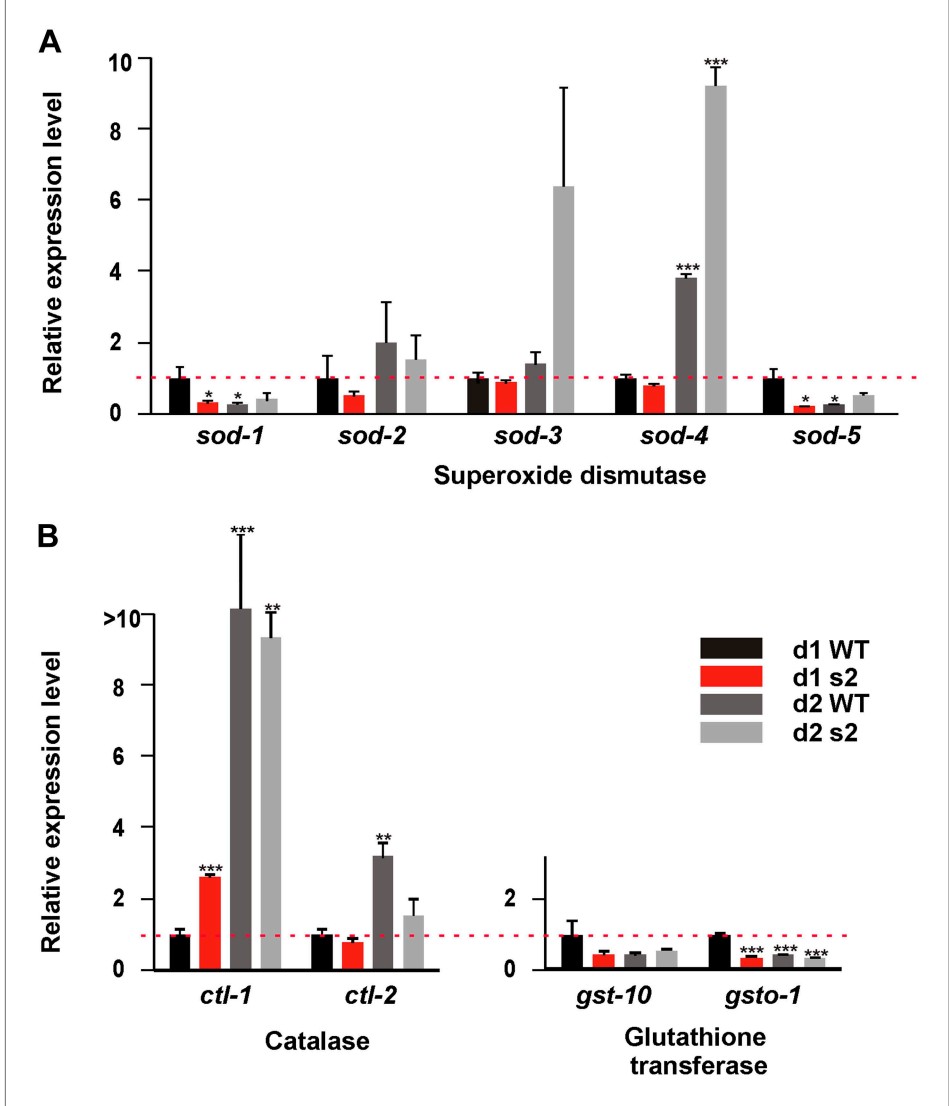

**Figure 7**. *sir-2.1(0)* males have compromised expression of anti-oxidant genes. Relative mRNA expression level of anti-oxidant genes superoxide dismutase (**A**), catalase (**B**), and glutathione transferase (**C**) in 1, 2-day-old wild type and *sir-2.1(0)* males (unpaired t-test).

adulthood day 3, and this decline is correlated with an increased excitability in the sex circuitry (**Guo et al., 2012**). Similar to aging males, a recent study using hermaphrodites showed that synaptic transmission in the locomotion circuit is enhanced after 5 days of adulthood (**Mulcahy et al., 2013**). Thus, it is possible that during early aging, the neuromuscular systems of both sexes become hyperexcitable. However, the faster decline in mating suggests that the male reproductive circuitry is more sensitive to age-related physiological changes.

We found that SIR-2.1 is a modulator of behavior and is required to maintain mating in aging males. 1-day-old *sir-2.1(0)* males can mate similarly to wild type, suggesting that *sir-2.1* is not essential for mating. However, unlike wild-type males, the mating ability of *sir-2.1(0)* prematurely drops at day 2. This is due to hyperexcitability of the reproductive circuitry that coordinates spicule intromission and ejaculation. The hyperexcitability of the spicule muscles causes the male proctodeum to block the connection between the vas deferens and the cloacal opening, which indirectly obstructs the transfer of sperm. The mutant phenotype resembles the behavioral, physiological, and pharmacological changes that occur in older wild-type males. This indicates that in wild type, SIR-2.1 maintains the functional excitability of the intromission and ejaculation circuit, possibly by slowing down the

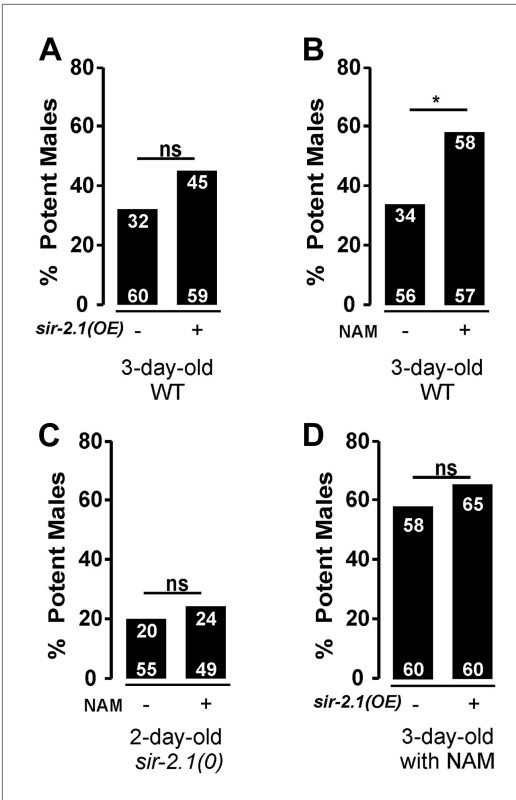

**Figure 8**. Exogenous nicotinamide improves mating during aging. *sir-2.1* overexpression cannot increase mating potency of 3-day-old wild type (**A**). However, feeding with a NAD$^+$ precursor nicotinamide (NAM) significantly improve the mating potency of 3-day-old wild type (**B**) but not 2-day-old *sir-2.1(0)* males (**C**). Overexpression of *sir-2.1* cannot further promote the effect of exogenous NAM (**D**).

deteriorative events that accumulate during aging. We suggest that as the male ages, SIR-2.1 regulates the amounts of catabolic, anabolic, and free radical scavenging enzymes to balance the energy demands needed for rapid reproductive motor responses, with the generation of damaging metabolic by-products, such as ROS.

Our data indicate that the male's cellular physiology is correlated with his ability to mate. We showed previously and in this study that perturbation of the male's physiology through genetic mutation or through dietary alterations during early adulthood will affect his ability to mate later in life. The physiology of wild-type males is likely changing from day 1 to day 2, as determined by the level of mRNAs encoding metabolic enzymes and the amount of terminal metabolic products. These physiological changes can ultimately lead to excessive carbon flow into the TCA cycle, and consequently, more NADH to be oxidized by ETC complexes (**Figures 6A and 9**). This promotes ROS generation via electron leak (**Federico et al., 2012**), which can be reflected by behavioral decay at day 3. Analysis of *sir-2.1(0)* males allowed us to extrapolate how this protein deacetylase regulates male physiology. Lack of SIR-2.1 will induce these deleterious changes to occur sooner, and degenerative behavioral responses in the mutant males are measured at day 2. Under standard laboratory conditions, wild-type males are raised constitutively on abundant *E. coli* until senescence. We speculate that since SIR-2.1 uses NAD$^+$ as a cofactor to deacetylate proteins, the physiological changes that occur after 2 days of constitutive feeding in wild type adults might be due to reduction in SIR-2.1 function, via the lower

ratio of NAD$^+$ to NADH in wild type. The phenomenon of altering NAD$^+$ to NADH levels in vertebrate cells is shown to reduce the activity of SIRT1 (**Braidy et al., 2011**). Decreased activity of SIR-2.1 will not only aggravate a bias towards catabolism, but will also decrease the levels of ROS scavengers. This is consistent with a hermaphrodite study, which showed that *sir-2.1* overexpression can protect the organism from ROS, possibly via HCF-1 and FOXO/DAF-16, to regulate the expression of stress response genes (**Rizki et al., 2011**).

Although our qPCR analyses suggest that enhanced catabolism might be occurring in 1-day-old of *sir-2.1(0)* and 2-day-old wild-type males, their mating ability could be facilitated via anabolic compensatory mechanisms. In addition to enhanced expression of catabolic genes, mRNAs encoding enzymes such as pyruvate carboxylase and phosphoenolpyruvate carboxykinase (PEPCK) appeared to be also up-regulated. This could be a likely reason for why the males contain more measurable lipids and glycogen. We propose that the up-regulation of anabolic process is a self-compensatory mechanism to divert carbon from the TCA cycle (**Figure 9**). *sir-2.1(0)* males that are mutant for PEPCK genes, lose their ability to generate fat and fail to mate efficiently on day 1. Likewise *sir-2.1 (+)* males with a mutation in PEPCK genes also display premature mating decline. We hypothesize that anabolic pathways could act as a homeostatic mechanism to reduce ROS production. This idea raises the possibility that obesity as a phenotype might be a compensatory mechanism to alleviate the effects of other underlying metabolic dysfunctions. Indeed, by switching of glycolysis to gluconeogenesis has been shown to be a potential strategy to cure hepatocarcinoma (**Ma et al., 2013**). A recent study showed that diabetic patients, who are not obese, have higher mortality than overweight ones (**Carnethon et al., 2012**).

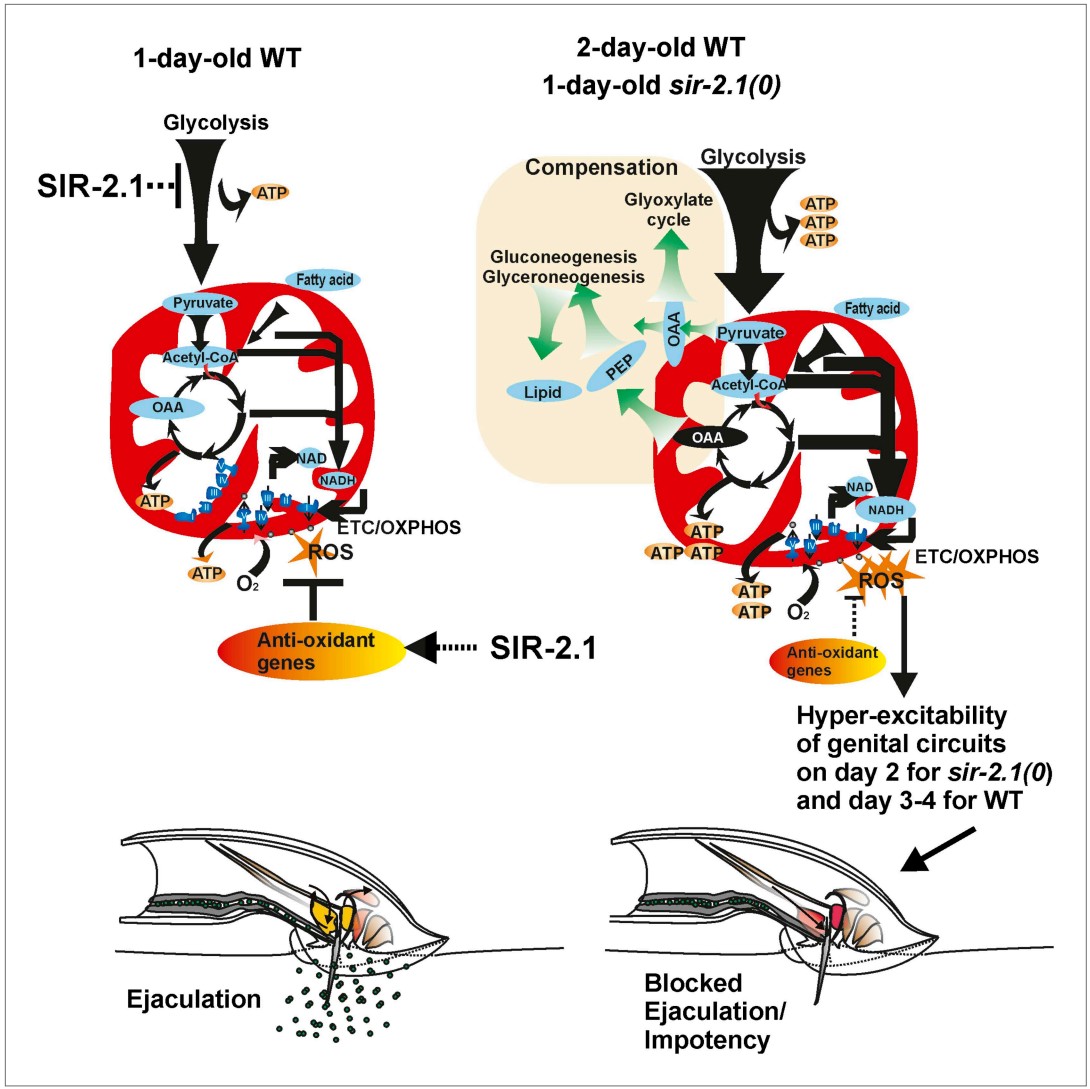

**Figure 9**. A cartoon of the metabolism and behavior that occurs in wild-type and *sir-2.1(0)* males during early aging. For successful reproductive behavior, SIR-2.1 is required to maintain proper carbon flow to meet the male's energy demands and balance the generation of ROS. In 1-day-old old *sir-2.1(0)* males, catabolism such as glycolysis and fatty acid oxidation is enhanced, and consequently, oxidative phosphorylation and generation of ROS are also increased. Without SIR-2.1, ROS accumulation by day 2 of adulthood can lead to hyperexcitability of the male's genital neuromuscular circuitry. This results in blocked ejaculation and impotency. It is possible that in 2- to 3-day-old wild-type males, the NAD+-dependent SIR-2.1 activity declines due to a lower ratio of NAD to NADH; thus older wild-type males might have a similar physiology as 1-day-old *sir-2.1(0)* males.

Another study showed that a lifestyle intervention focusing on weight loss did not reduce the rate of cardiovascular events in obese adults with type II diabetes (*Look AHEAD Research Group, 2013*), challenging the traditional viewpoint that obesity is a major contributor to metabolic disorders.

The free radical aging theory states that aging is the result of free radical-induced molecular damages (*Harman, 1956*). Although this notion is challenged by the hormesis theory, which posits that moderate amounts of physiological or environmental insults can reinforce cellular processes that reduce stress-induced damage (*Afanas'ev, 2010*; *Schulz et al., 2007*), artificially applied ROS is reported to change the excitability of cultured neurons and muscles through chemically damaging ion channels (*Danson and Paterson, 2006*). Therefore, aspects of the free radical theory of aging might still apply to physiological changes in neural muscular systems and behavioral decay. Consistent with this, we showed here that genetic and environmental conditions, which can lead to

oxidative stress, caused increased excitability of the male mating circuitry and behavioral decay during early aging.

There is not a strict correlation between oxidative stress and changes in the electrical properties of neurons and muscles. Oxidation of different types of ion channels changes their conductance and alters the cell's excitability (*Annunziato et al., 2002*). One *C. elegans* study indicates that oxidative stress reduces cell excitability by increasing the conductance of K+ channels (*Sesti et al., 2010*). Another mammalian study showed that oxidative stress hyperpolarizes the resting potential, but extends the duration of the action potential in cardiac ganglion (*Whyte et al., 2009*). Mitochondria ROS has been shown to trigger $Ca^{2+}$ increases in the pulmonary arterial myocytes (*Waypa et al., 2002*, *2006*). In a study using glia, L-type voltage-gated $Ca^{2+}$ channels (L-VGCC) were found to be a target of ROS. After modification by ROS, their conductance of $Ca^{2+}$ was increased (*Bond and Greenfield, 2007*). Different from vertebrate skeletal muscles, L-VGCC in *C. elegans* propagates action potentials, and the entry of external $Ca^{2+}$ directly promotes excitation–contraction coupling (*Lee et al., 1997*; *Maryon et al., 1998*). Previous work showed that L-VGCC EGL-19 in *C. elegans* is required for sustained tonic contraction of the copulatory spicule muscles (*Garcia et al., 2001*). Therefore, oxidation of L-VGCC in *C. elegans* might contribute to the increased excitability of mating circuits. Other major targets of ROS are voltage-gated K+ channels. In *C. elegans*, oxidation of KVS-1 slows down its inactivation, leading to hyperpolarization and sensory function loss (*Cai and Sesti, 2009*). The ERG-like K+ channel UNC-103 is a major excitability regulator of the sex circuit (*Reiner et al., 2006*). Although there is no report of oxidative modification on UNC-103, human encoded H-ERG channels can be activated by oxidative stress, so that cells become hyperpolarized and less excitable (*Cui and Zhang, 2013*). Considering that ROS increases the excitability of the mating circuit, it is possible that L-VGCC is more prone to be oxidized than K+ channels in male reproductive cells.

## Materials and methods

### Strains and culture conditions

Worms were grown at 20°C on nematode growth media (NGM) plates seeded with *E. coli* strain OP50, except for strains containing the *pha-1(e2123)*, which were maintained at 15°C. The alleles used in this work included: *lite-1(ce314)* on LGX; *pck-2(ok2586)* on LGI; *pck-1(ok2098)*, *pha-1(e2123)* (*Schnabel and Schnabel, 1990*), *unc-64(e240)* (*Brenner, 1974*) on LGIII; *sir-2.1(ok434)* on LGIV; and *him-5(e1490)* (*Hodgkin et al., 1979*) on LGV. The males were generated by the *him-5(e1490)* mutation. Males containing only this mutation are referred to as wild-type; *him-5(e1490)* males have been shown to mate efficiently as wild type (*Hodgkin, 1983*). *pck-2(ok2586)*, *pck-1(ok2098)*, and *sir-2.1(ok434)* were generated by the *C. elegans* Gene Knockout Consortium. *sir-2.1(ok434)* animals were out-crossed 4 times with the *him-5(e1490)* strain. The deletion in *sir-2.1(0)* was detected through PCR using primers listed in the *Supplementary file 2*.

Altered mediums used here included NGM medium containing glucose (2%), paraquat, N-acetyl-cystine (NAC) (Sigma, St. Louis, MO), and nicotinamide (NAM) (Sigma) respectively. The latter three were added at the indicated concentration just before pouring the plates. OP50 used for the special medium containing glucose and NAC was UV-killed and concentrated to make sure the worms were not food deprived. We assume that the animals ingest these compounds as they feed on *E. coli* or absorb them through their cuticle.

### Transgenic constructs

DNA primers are listed in the *Supplementary file 2*. The *sir-2.1* genomic sequence, plus 2 kb upstream of its ATG, was PCR-amplified from N2 DNA. The PCR product was digested and ligated between the *SphI* and *SalI* sites of pSX322YFP to obtain the plasmid pXG5. To obtain promoters for driving *sir-2.1* expression, the *sir-2.1* endogenous promoter was removed via PCR-mutagenesis from pXG5 to construct pXG6. The Gateway ATTR cassette frame A was inserted in front of the *sir-2.1* genomic sequence to make the destination clone pXG7. Plasmids (pXG8, pXG9, and pXG11) that promote neuronal, muscular, and intestinal expression of *sir-2.1* were obtained through Gateway LR reactions between pXG7 and pLR35 (*Paex-3*) (*LeBoeuf et al., 2007*), pLR22 (*Plev-11*) (*Gruninger et al., 2008*) and pBL50 (*Pges-1*) (*Urano et al., 2002*), respectively. pXG5 (25 ng/µl), pXG8 (10 ng/µl), pXG9 (1 ng/µl) or pXG11 (50 ng/µl) were injected to *sir-2.1(0)* hermaphrodites and transgenic animals were selected

via YFP fluorescence. pXG5 (50 ng/µl) was injected into wild-type hermaphrodites to obtain strains with overexpression of *sir-2.1* (referred as *sir-2.1(OE)*).

## Behavioral assays

The mating potency assay was performed as described in *Guo et al. (2012)*. Briefly, L4 males were isolated from hermaphrodites. One male and one *pha-1(e2123)* hermaphrodite were co-transferred to a 5-mm-diameter OP50 mating lawn at 20°C, which is a restrictive temperature for *pha-1(e2123)* to sire viable self-progeny. This allowed us to score the mating potency of males, as only cross-progenies would develop to adulthood.

To quantify different parameters of mating, we observed, up to 5 min, copulations between males and paralyzed *unc-64(e240)* hermaphrodites. The ability of males to sense the vulva was calculated by counting the number of times he stopped at the vulva divided by the total number of times he stopped and/or passed by the vulva. The turning quality was calculated as: the number of smooth turns (defined as the male tail keeping contact with hermaphrodite and turning without hesitation) divided by the total number of turns. Ejaculation assays were conducted in two ways. We directly observed sperm transfer after spicule insertion, and determined if sperm drained into the *unc-64(e240)* hermaphrodite's uterus. Additionally, cross-progeny were counted 1–2 days after spicules insertion.

## Lifespan and stress resistance assays

Lifespan assay was conducted as described in *Guo et al. (2012)*. L4 males were isolated and raised 20 per plate. The males that can respond to gentle touch with a platinum wire were counted and transferred to new plates every day. Males that dried on the wall of the Petri plate were censored from the assay on the day they died.

To assess the sensitivity to paraquat, L4 males were transferred to plates containing 10 mM paraquat and scored at 24 and 48 hr.

## In vitro sperm activation assay

Sperm activation assays were done according to *L'Hernault and Roberts (1995)*; *Smith and Stanfield (2011)*. Briefly, three 2-day-old males, isolated at L4 stage, were cut at the posterior portion with a needle in 20 µl sperm media (50 mM Hepes, pH7.0, 45 mM NaCl, 25 mM KCl, 1 mM $MgSO_4$, and 5 mM $CaCl_2$) freshly supplemented with polyvinylprolidone (PVP) 40,000 molecular weight (Sigma) and the activator pronase (Roche, Indianapolis, IN) at the final concentration of 10 mg/ml and 500 µg/ml on a slide. A coverslip with a thin layer of Vaseline applied around the edge was put on the top of the sperm media to form a chamber over the sperm. After 5-min incubation, activated sperm with a pseudopod and inactive ones were counted using a compound scope fitted with a 100X objective. Approximately, 50–60 sperm cells were counted in each sample section.

## Drug tests

Arecoline (ARE) and levamisole (LEV) were dissolved in water at the concentration of 100 mM, and then diluted accordingly, as described in *Liu et al. (2007)*. Briefly, 50 µM of ARE, 1 µM and 500 nM of LEV for 1, 2-day-old males, and 100 nM LEV for 3-day-old males were used. Males were introduced to 1 ml drug baths and scored if they protruded their spicules for more than 5 s within 5 min of drug exposure.

## Ca²⁺ imaging

$Ca^{2+}$ imaging was conducted and analyzed as described in *Guo et al. (2012)*. 50 ng/µl pLR289[P*unc-103E:G-CaMP3::sl2:::DsRed*] (*Correa et al., 2012*) and 50 ng/µl pBX1(*Granato et al., 1994*) were co-injected into *pha-1(e2123); him-5(e1490); lite-1(ce314)* hermaphrodites to generate the extrachromosomal array *rgEx566[Punc-103E:G-CaMP3::sl2:::DsRed]*. *rgEx566* was crossed into *pha-1(e2123); him-5(e1490); sir-2.1(0); lite-1(ce314)* hermaphrodites. Fluorescence signals from G-CaMP3 and DsRed were recorded simultaneously during the copulations of 2-day-old wild-type and *sir-2.1(0)* males with paralyzed hermaphrodites containing the transgene *rgEx431[Phsp-16:egl-2(gf)*; *Punc-103E:DsRed]*.

## Real-time PCR

300 day 1 and day 2 adult males were accumulated over a period of time. RNA was extracted by Trizol, and cDNA was synthesized by SuperScript II (Life technology, Grand Island, NY) using around 2 µg total RNA, as described in *LeBoeuf and Garcia (2012)*. The RT-qPCR reactions were performed using

BIO-RAD CFX96 real-time system and SsoFast EvaGreen supermix. 11 candidate reference genes were tested to see whether their expression changed from day 1 and day 2 in both wild-type and *sir-2.1* males (*Hoogewijs et al., 2008*); from our analyses, *act-1* and *gpd-2* were selected as the reference to normalize the expression of the metabolic genes. Many of the primers used to detect the expression of metabolic enzymes are described in *Castelein et al. (2008)*. Other primers for additional metabolic and antioxidant stress genes are listed in the *Supplementary file 2*. Three replicates were conducted on the same RNA samples. We used the t-test to determine which mRNA transcripts in *sir-2.1*(0) males were significantly different from their cognate wild-type transcripts.

### ATP, glucose, glycogen, and lipid measurements

To measure ATP and glucose, 100 males were collected at different ages, frozen and thawed three times. The worms were homogenized, and the supernatant was collected and measured using an ATP determination Kit (Life technology) and the Glucose Oxidase Assay Kit (Life technology). The ATP and glucose were normalized to the amount of dsDNA quantified by picoGreen (Life technology).

To stain glycogen, 1-day-old *sir-2.1(0)* and wild-type virgin males were transferred to 2% agar pads. The pads containing both genotypes were then placed over a bottle of iodine crystals for 30 s (*Frazier and Roth, 2009*). The pictures were taken by a *Leica* compound scope mounted with OLYMPUS DP70 camera. The RGB images were then converted to 16-bit gray scale, and the mean gray levels of the isthmus regions were measured using the SimplePCI image quantification software (Hamamatsu, Janpan). The mean gray level was reversely correlated with the red signal.

Oil Red O staining was done according to *O'Rourke et al. (2009)*. Briefly, males were collected and washed with PBS, and then fixed with Modified Ruvkuns witches brew (MRWB) buffer containing 1% paraformaldehyde (PFA) for 1 hr. Worms were then washed with PBS and suspended in 60% isopropanol for 15 min at room temperature. The 60% isopropanol was removed and worms were bathed in 60% Red Oil O staining solution. The RGB images were taken by a *Leica* compound scope mounted with OLYMPUS DP70 camera. The images were quantified by ImageJ according to *Mehlem et al. (2013)*.

## Acknowledgements

We thank D Gualberto for technical assistance, and B LeBoeuf, C Jee, P Correa, L Zhang, Y Liu, and X Chen for discussion of the project and critical reading of the manuscript. *C. elegans* strains were provided by the *Caenorhabditis* Genetic Center. This work is supported by the Howard Hughes Medical Institute.

## Additional information

### Funding

| Funder | Author |
| --- | --- |
| Howard Hughes Medical Institute | L René García |

The funder had no role in study design, data collection and interpretation, or the decision to submit the work for publication.

### Author contributions

XG, Conception and design, Acquisition of data, Analysis and interpretation of data, Drafting or revising the article; LRG, Conception and design, Analysis and interpretation of data, Drafting or revising the article

## Additional files

### Supplementary files

• Supplementary file 1. Expression levels of metabolic enzymes though real-time PCR. d1 WT refers to day1 wild type; d2 WT refers to day 2 wild type; d1 s2 refers to day1 *sir-2.1(0)*; d2 s2 refers to day 2 *sir-2.1(0)*.

• Supplementary file 2. Primers used in this study.

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
