## [Decision Letter]

Thank you for sending your work entitled “SIR-2.1 integrates metabolic homeostasis with the reproductive neuromuscular excitability in aging male *C. elegans*” for consideration at *eLife*. Your article has been favorably evaluated by a Senior editor and three reviewers, one of whom served as a guest Reviewing editor.

The guest Reviewing editor and the other reviewers discussed their comments before we reached this decision, and the Reviewing editor has assembled the following comments to help you prepare a revised submission.

The reviewers found the work and its novelty to be of the highest level. However, two concerns were raised.

1) The need to test *sir-2.1* overexpression phenotypes to ask if *sir-2.1* is sufficient for the described phenotypes.

2) The need to better quantify the metabolic phenotypes suggested in the manuscript. All three reviewers realize that this might be technically challenging to perform metabolic analysis on populations of male worms, but also believe that the qPCR data can be expanded and more tightly controlled. Because qPCR will be proxy for actual metabolic profiling, the reviewers all indicate that the claims to altered metabolism must be tempered in the manuscript.

---

## [Author Response]

*1) The need to test sir-2.1 overexpression phenotypes to ask if sir-2.1 is sufficient for the described phenotypes*.

So this can be distilled to the question, “Does overexpression of *sir-2.1* prolong mating?”

The quick answer is no. It has been implicated that overexpression of *sir-2.1* extends lifespan, although still controversial, it would be of interest to determine whether overexpression of *sir-2.1* could also improve the mating ability of 3 day-old wild type.

To test this, we conducted the mating potency assay with 3 day-old wild-type males containing an extra chromosome array of *sir-2.1* fused to *yfp*, driven by *sir-2.1’s* promoter. Compared to the non-transgenic wild type, the overexpression of *sir-2.1* does not increase the mating potency. This is not due to *sir-2.1* fused to *yfp*, since the same plasmid can rescue the *sir-2.1(0)* phenotype.

Another way to boost *sir-2.1* activity is to supply worms with the NAD^+^ precursor such as nicotinamide (NAM) (32). Thus we thought we should also explore this possibility before we rule out that more *sir-2.1* expression does not extend wild-type mating. After supplying worms with 200 µM NAM, we found that it did improve mating of 3 day-old wild type males but not 2 day-old *sir-2.1(0)* males, indicating that NAM supplementation improves mating in a *sir-2.1* dependent manner.

However, overexpression of *sir-2.1* does not further amplify the positive effects of NAM feeding. This indicates that during early aging, NAM or possibly NAD^+^ additionally promotes behavioral extension through mechanisms parallel to SIR-2.1 activity.

We thank the reviewers for suggesting these experiments since it adds an additional dimension to our manuscript. These results are included in a new Results section entitled “Nicotinamide delays the deterioration of male mating behavior”.

*2) The need to better quantify the metabolic phenotypes suggested in the manuscript. All 3 reviewers realize that this might be technically challenging to perform metabolic analysis on populations of male worms, but also believe that the qPCR data can be expanded and more tightly controlled. Because qPCR will be proxy for actual metabolic profiling, the Reviewers all indicate that the claims to altered metabolism must be tempered in the manuscript*.

We are currently having metabolic profiling done by the company Metabolon, of ∼200 metabolites on day 1, day 2, and day 3 age-synchronized males (in total ∼1.5 million wild type and *sir-2.1(0)*); but the company tells us that they would not be able to provide us the data until April. We were hoping that we could get the data sooner for this resubmission, but it does not look possible. We hope that the reviewers would understand the time constraints involved in the experiment and analyses.

In our first submission of the qPCR experiment, we did not fully explain the extent of the qPCR analysis or the statistical comparisons to determine whether changes in expression levels were significant. We apologize for the omission, and have included the data in a more accessible form. This information is included in the Materials and methods, as well as the text.

In order to predict the metabolic state of *sir-2.1(0)* and wild type during early aging, we hand-picked 1200 age-matched males over two age stages and determined the expression of 55 metabolic genes. We tested 11 potential reference genes to determine which two were the appropriate reference genes that did not change in both wild-type and *sir-2.1(0)* males over the two age stages. The expression information of all 55 genes is list in the [Supplementary-material SD1-data]. We then used the t-test to identify which gene expression levels in *sir-2.1(0)* were significantly different compared to their cognate wild-type reference. Out of 55 genes, 17 showed significant differences with a *p* value of <0.05. To better display the data, we re-organized the expression of those 17 genes from a table form to a bar form, with significance asterisks added.

Consistent with the following reports from the literature: the key enzyme of glycolysis hexokinase is up-regulated in SIRT1(SIR-2.1 ortholog) iKO mice (22); Hepatic *Sirt1* deficiency cause increased expression of gluconeogenesis enzyme phosphoenolpyruvate carboxykinase (PEPCK, equivalent to *pck-1* and *pck-2* in *C. elegans*) (70), we also detected up-regulation of hexokinase and PEPCK in our qPCR data. Thus some of our qPCR results complement the published reports.

Although this gave us the confidence on the qPCR results, we are aware that the expression level may not truly reflect the final activity, therefore we determined the end products of those metabolic pathways, reasoning that the intermediates may not significantly change because of the reversibility of the metabolic pathways, for example, glycolysis and gluconeogenesis share most of the intermediates. We found that changes in lipids, glycogen and ATP were consistent with the qPCR results.

However, as suggested by the reviewers, we toned down our statements in the Results and Discussion that metabolism is altered in the *sir-2.1(0)* males.